# Extracting Certainty from Uncertainty: Transductive Pairwise Classification from Pairwise Similarities

**Tianbao Yang[†], Rong Jin[‡♮]**
[†]The University of Iowa, Iowa City, IA 52242
[‡]Michigan State University, East Lansing, MI 48824
[♮]Alibaba Group, Hangzhou 311121, China
`tianbao-yang@uiowa.edu, rongjin@msu.edu`

## Abstract

In this work, we study the problem of transductive pairwise classification from pairwise similarities [1]. The goal of transductive pairwise classification from pairwise similarities is to infer the pairwise class relationships, to which we refer as pairwise labels, between all examples given a subset of class relationships for a small set of examples, to which we refer as labeled examples. We propose a very simple yet effective algorithm that consists of two simple steps: the first step is to complete the sub-matrix corresponding to the labeled examples and the second step is to reconstruct the label matrix from the completed sub-matrix and the provided similarity matrix. Our analysis exhibits that under several mild preconditions we can recover the label matrix with a small error, if the top eigen-space that corresponds to the largest eigenvalues of the similarity matrix covers well the column space of label matrix and is subject to a low coherence, and the number of observed pairwise labels is sufficiently enough. We demonstrate the effectiveness of the proposed algorithm by several experiments.

## 1 Introduction

Pairwise classification aims to determine if two examples belong to the same class. It has been studied in several different contexts, depending on what prior information is provided. In this paper, we tackle the pairwise classification problem provided with a pairwise similarity matrix and a small set of true pairwise labels. We refer to the problem as **transductive pairwise classification from pairwise similarities**. The problem has many applications in real world situations. For example, in network science [17], an interesting task is to predict whether a link between two nodes is likely to occur given a snapshot of a network and certain similarities between the nodes. In computational biology [16], an important problem is to predict whether two protein sequences belong to the same family based on their sequence similarities, with some partial knowledge about protein families available. In computer vision, a good application can been found in face verification [5], which aims to verify whether two face images belong to the same identity given some pairs of training images.

The challenge in solving the problem arises from the uncertainty of the given pairwise similarities in reflecting the pairwise labels. Therefore the naive approach by binarizing the similarity values with a threshold would suffer from a bad performance. One common approach towards the problem is to cast the problem into a clustering problem and derive the pairwise labels from the clustering results. Many algorithms have been proposed to cluster the data using the pairwise similarities and a subset of pairwise labels. However, the success of these algorithms usually depends on how many pairwise labels are provided and how well the pairwise similarities reflect the true pairwise labels as well.

In this paper, we focus on the theoretical analysis of the problem. Essentially, we answer the question of what property the similarity matrix should satisfy and how many pre-determined pairwise labels are sufficient in order to recover the true pairwise labels between all examples. We base our analysis on a very simple scheme which is composed of two steps: (i) the first step recovers the sub-matrix of the label matrix from the pre-determined entries by matrix completion, which has been studied extensively and can be solved efficiently; (ii) the second step estimates the full label matrix by simple matrix products based on the top eigen-space of the similarity matrix and the completed sub-matrix. Our empirical studies demonstrate that the proposed algorithm could be effective than spectral clustering and kernel alignment approach in exploring the pre-determined labels and the provided similarities.

To summarize our theoretical results: under some appropriate pre-conditions, namely the distribution of data over the underlying classes in hindsight is well balanced, the labeled data are uniformly sampled from all data and the pre-determined pairwise labels are uniformly sampled from all pairs between the labeled examples, we can recover the label matrix with a small error if (i) the top eigen-space that corresponds to the $s$ largest eigen-values of the similarity matrix covers well the column space of the label matrix and has a low incoherence, and (ii) the number of pre-determined pairwise labels $N$ on $m$ labeled examples satisfy $N \geq \Omega(m \log^2(m)), m \geq \Omega(\mu_s s \log s)$, where $\mu_s$ is a coherence measure of the top eigen-space of the similarity matrix.

## 2 Related Work

The transductive pairwise classification problem is closely related to semi-supervised clustering, where a set of pairwise labels are provided with pairwise similarities or feature vectors to cluster a set of data points. We focus our attention on the works where the pairwise similarities instead of the feature vectors are served as inputs.

Spectral clustering [19] and kernel k-means [7] are probably the most widely applied clustering algorithms given a similarity matrix or a kernel matrix. In spectral clustering, one first computes the top eigen-vectors of a similarity matrix (or bottom eigen-vectors of a Laplacian matrix), and then cluster the eigen-matrix into a pre-defined number of clusters. Kernel k-means is a variant of k-means that computes the distances using the kernel similarities. One can easily derive the pairwise labels from the clustering results by assuming that if two data points assigned to the same cluster belong to the same class and vice versa. To utilize some pre-determined pairwise labels, one can normalize the similarities and replace the entries corresponding to the observed pairs with the provided labels.

There also exist some works that try to learn a parametric or non-parametric kernel from the pre-determined pairwise labels and the pairwise similarities. Hoi et al. [13] proposed to learn a parametric kernel that is characterized by a combination of the top eigen-vectors of a (kernel) similarity matrix by maximizing a kernel alignment measure over the combination weights. Other works [2, 6] that exploit the pairwise labels for clustering are conducted using feature vector representations of data points. However, all of these works are lack of analysis of algorithms, which is important from a theoretical point. There also exist a large body of research on preference learning and ranking in semi-supervised or transductive setting [1, 14]. We did not compare with them because that the ground-truth we analyzed of a pair of data denoted by $h(u, v)$ is a symmetric function, i.e., $h(u, v) = h(v, u)$, while in preference learning the function $h(u, v)$ is an asymmetric function.

Our theoretical analysis is built on several previous studies on matrix completion and matrix reconstruction by random sampling. Candès and Recht [3] cooked a theory of matrix completion from partial observations that provides a theoretical guarantee of perfect recovery of a low rank matrix under appropriate conditions on the matrix and the number of observations. Several works [23, 10, 15, 28] analyzed the approximation error of the Nyström method that approximates a kernel matrix by sampling a small number of columns. All of these analyses exploit an important measure of an orthogonal matrix, i.e., matrix incoherence, which also plays an important role in our analysis.

It has been brought to our attention that two recent works [29, 26] are closely related to the present work but with remarkable differences. Both works present a matrix completion theory with side information. Yi et al. [29] aim to complete the pairwise label matrix given partially observed entries for semi-supervised clustering. Under the assumption that the column space of the symmetric

pairwise label matrix to be completed is spanned by the top left singular vectors of the data matrix, they show that their algorithm can perfectly recover the pairwise label matrix with a high probability. In [26], the authors assume that the column and row space of the matrix to be completed is given aprior and show that the required number of observations in order to perfectly complete the matrix can be reduced substantially. There are two remarkable differences between [29, 26] and our work: (i) we target on a transductive setting, in which the observed partial entries are not uniformly sampled from the whole matrix; therefore their algorithms are not applicable; (ii) we prove a small reconstruction error when the assumption that the column space of the pairwise label matrix is spanned by the top eigen-vectors of the pairwise similarity matrix fails.

# 3 The Problem and A Simple Algorithm

We first describe the problem of transductive pairwise classification from pairwise similarities, and then present a simple algorithm.

## 3.1 Problem Definition

Let $\mathcal{D}_n = \{o_1, \ldots, o_n\}$ be a set of $n$ examples. We are given a pairwise similarity matrix denoted by $S \in \mathbb{R}^{n \times n}$ with each entry $S_{ij}$ measuring the similarity between $o_i$ and $o_j$, a set of $m$ random samples denote by $\widehat{\mathcal{D}}_m = \{\hat{o}_1, \ldots, \hat{o}_m\} \subseteq \mathcal{D}_n$, and a subset of pre-determined pairwise labels being either 1 or 0 that are randomly sampled from all pairs between the examples in $\widehat{\mathcal{D}}_m$. The problem is to recover the pairwise labels of all remaining pairs between examples in $\mathcal{D}_n$. Note that the key difference between our problem and previous matrix completion problems is that the partial observed entries are only randomly distributed over $\widehat{\mathcal{D}}_m \times \widehat{\mathcal{D}}_m$ instead of $\mathcal{D}_n \times \mathcal{D}_n$.

We are interested in that the pairwise labels indicate the pairwise class relationships, i.e., the pairwise label between two examples being equal to 1 indicates they belong to the same class, and being equal to 0 indicates that they belong to different classes. We denote by $r$ the number of underlying classes. We introduce a label matrix $Z \in \{0,1\}^{n \times n}$ to represent the pairwise labels between all examples, and similarly denote by $\widehat{Z} \in \{0,1\}^{m \times m}$ the pairwise labels between any two labeled examples [2] in $\widehat{\mathcal{D}}_m$. To capture the subset of pre-determined pairwise labels for the labeled data, we introduce a set $\Sigma \subset [m] \times [m]$ to indicate the subset of observed entries in $\widehat{Z}$, i.e., the pairwise label $\widehat{Z}_{i,j}, (i,j) \in \Sigma$ is observed if and only if the pairwise label between $\hat{o}_i$ and $\hat{o}_j$ is pre-determined. We denote by $\widehat{Z}_\Sigma$ the partially observed label matrix, i.e.

$$[\widehat{Z}_\Sigma]_{i,j} = \left\{ \begin{array}{ll} \widehat{Z}_{i,j} & (i,j) \in \Sigma \\ \text{N\textbackslash A} & (i,j) \notin \Sigma \end{array} \right.$$

The goal of **transductive pairwise classification from pairwise similarities** is to estimate the pairwise label matrix $Z \in \{0,1\}^{n \times n}$ for all examples in $\mathcal{D}_n$ using (i) the pairwise similarities in $S$ and (ii) the partially observed label matrix $\widehat{Z}_\Sigma$.

## 3.2 A Simple Algorithm

In order to estimate the label matrix $Z$, the proposed algorithm consists of two steps. The first step is to recover the sub-matrix $\widehat{Z}$, and the second step is to estimate the label matrix $Z$ using the recovered $\widehat{Z}$ and the provided similarity matrix $S$.

**Recover the sub-matrix $\widehat{Z}$** First, we note that the label matrix $Z$ and the sub-matrix $\widehat{Z}$ are of low rank by assuming that the number of hidden classes $r$ is small. To see this, we let $\mathbf{g}_k \in \{1,0\}^n, \widehat{\mathbf{g}}_k \in \{1,0\}^m$ denote the class assignments to the $k$-th hidden class of all data and the labeled data, respectively. It is straightforward to show that

$$Z = \sum_{k=1}^{r} \mathbf{g}_k \mathbf{g}_k^\top, \quad \widehat{Z} = \sum_{k=1}^{r} \widehat{\mathbf{g}}_k \widehat{\mathbf{g}}_k^\top \tag{1}$$

**Algorithm 1** A Simple Algorithm for Transductive Pairwise Classification by Matrix Completion

---
1: **Input:**
- $S$: a pairwise similarity matrix between all examples in $\mathcal{D}_n$
- $\widehat{Z}_\Sigma$: the subset of observed pairwise labels for labeled examples in $\widehat{\mathcal{D}}_m$
- $s < m$: the number of eigenvectors used for estimating $Z$

2: Compute the first $s$ eigen-vectors of a similarity matrix $S$ // *Preparation*

3: Estimate $\widehat{Z}$ by solving the optimization problem in (2) // *Step 1: recover the sub-matrix $\widehat{Z}$*

4: Estimate the label matrix $Z$ using (5) // *Step 2: estimate the label matrix $Z$*

5: **Output:** $Z$

---

which clearly indicates that both $Z, \widehat{Z}$ are of low rank if $r$ is significantly smaller than $m$. As a result, we can apply the matrix completion algorithm [20] to recover $\widehat{Z}$ by solving the following optimization problem:

$$\min_{M \in \mathbb{R}^{m \times m}} \|M\|_{tr}, \quad \text{s.t.} \quad M_{i,j} = \widehat{Z}_{i,j} \ \forall (i,j) \in \Sigma \tag{2}$$

where $\|M\|_{tr}$ denotes the nuclear norm of a matrix.

**Estimate the label matrix $Z$** The second step is to estimate the remaining entries in the label matrix $Z$. In the sequel, for the ease of analysis, we will attain an estimate of the full matrix $Z$, from which one can obtain the pairwise labels between all remaining pairs.

We first describe the motivation of the second step and then present the details of computation. Assuming that there exists an orthogonal matrix $U_s = (\mathbf{u}_1, \cdots, \mathbf{u}_s) \in \mathbb{R}^{n \times s}$ whose column space subsumes the column space of the label matrix $Z$ where $s \geq r$, then there exist $\mathbf{a}_k \in \mathbb{R}^s, k = 1, \ldots, r$ such that

$$\mathbf{g}_k = U_s \mathbf{a}_k, \quad k = 1, \ldots, r. \tag{3}$$

Considering the formulation of $Z$ and $\widehat{Z}$ in (1), the second step works as follows: we first compute an estimate of $\sum_{k=1}^r \mathbf{a}_k \mathbf{a}_k^\top$ from the completed sub-matrix $\widehat{Z}$, then compute an estimate of $Z$ based on the estimate of $\sum_{k=1}^r \mathbf{a}_k \mathbf{a}_k^\top$. To this end, we construct the following optimization problems for $k = 1, \ldots, r$:

$$\widehat{\mathbf{a}}_k = \arg\min \|\widehat{\mathbf{g}}_k - \widehat{U}_s \mathbf{a}\|_2^2 = (\widehat{U}_s^\top \widehat{U}_s)^\dagger \widehat{U}_s^\top \widehat{\mathbf{g}}_k \tag{4}$$

where $\widehat{U}_s \in \mathbb{R}^{m \times s}$ is a sub-matrix of $U_s \in \mathbb{R}^{n \times s}$ with the row indices corresponding to the global indices of the labeled examples in $\widehat{\mathcal{D}}_m$ with respect to $\mathcal{D}_n$. Then we can estimate $\sum_{k=1}^r \mathbf{a}_k \mathbf{a}_k^\top$ and $Z$ by

$$\sum_{k=1}^r \mathbf{a}_k \mathbf{a}_k^\top = (\widehat{U}_s^\top \widehat{U}_s)^\dagger \widehat{U}_s^\top \sum_{k=1}^r \widehat{\mathbf{g}}_k \widehat{\mathbf{g}}_k^\top \widehat{U}_s (\widehat{U}_s^\top \widehat{U}_s)^\dagger = (\widehat{U}_s^\top \widehat{U}_s)^\dagger \widehat{U}_s^\top \widehat{Z} \widehat{U}_s (\widehat{U}_s^\top \widehat{U}_s)^\dagger$$

$$Z' = \sum_{k=1}^r \mathbf{g}_k \mathbf{g}_k^\top = U_s \left( \sum_{k=1}^r \mathbf{a}_k \mathbf{a}_k^\top \right) U_s^\top = U_s (\widehat{U}_s^\top \widehat{U}_s)^\dagger \widehat{U}_s^\top \widehat{Z} \widehat{U}_s (\widehat{U}_s^\top \widehat{U}_s)^\dagger U_s^\top \tag{5}$$

In oder to complete the algorithm, we need to answer how to construct the orthogonal matrix $U_s = (\mathbf{u}_1, \cdots, \mathbf{u}_s)$. Inspired by previous studies on spectral clustering [18, 19], we can construct $U_s$ as the first $s$ eigen-vectors that correspond to the $s$ largest eigen-values of the provided similarity matrix. A justification of the practice is that if the similarity graph induced by a similarity matrix has $r$ connected components, then the eigen-space of the similarity matrix corresponding to the $r$ largest eigen-values is spanned by the indicator vectors of the components. Ideally, if the similarity graph is equivalent to the label matrix $Z$, then the indicator vectors of connected components are exactly $\mathbf{g}_1, \cdots, \mathbf{g}_r$. Finally, we present the detailed step of the proposed algorithm in Algorithm 1.

**Remarks on the Algorithm** The performance of the proposed algorithm will reply on two factors. First, how accurate is the recovered the sub-matrix $\widehat{Z}$ by matrix completion. According to our later analysis, as long as the number of observed entries is sufficiently large (e.g., $|\Sigma| \geq \Omega(m \log^2 m)$), one can exactly recover the sub-matrix $\widehat{Z}$. Second, how well the top eigen-space of $S$ covers the

column space of the label matrix $Z$. As shown in section 4, if they are close enough, the estimated matrix of $Z$ has a small error provided the number of labeled examples $m$ is sufficiently large (e.g., $m \geq \Omega(\mu_s s \log s)$, where $\mu_s$ is a coherence measure of the top eigen-space of $S$.

It is interesting to compare the proposed algorithm to the spectral clustering algorithm [19] and the spectral kernel learning algorithm [13], since all three algorithms exploit the top eigen-vectors of a similarity matrix. The spectral clustering algorithm employes a k-means algorithm to cluster the top eigen-vector matrix. The spectral kernel learning algorithm optimizes a diagonal matrix $\Lambda = diag(\lambda_1, \cdots, \lambda_s)$ to learn a kernel matrix $K = U_s \Lambda U_s^\top$ by maximizing the kernel alignment with the pre-determined labels. In contrast, we estimate the pairwise label matrix by $Z' = U_s M U_s^\top$ where the matrix $M$ is learned from the recovered sub-matrix $\widehat{Z}$ and the provided similarity matrix $S$. The recovered sub-matrix $\widehat{Z}$ serves as supervised information and the similarity matrix $S$ serves as the input data for estimating the label matrix $Z$ (c.f. equation 4). It is the first step that explores the low rank structure of $\widehat{Z}$ we are able to gain more useful information for the estimation in the second step. In our experiments, we observe improved performance of the proposed algorithm compared with the spectral clustering and the spectral kernel learning algorithm.

## 4 Theoretical Results

In this section, we present theoretical results regarding the reconstruction error of the proposed algorithm, which essentially answer the question of what property the similarity matrix should satisfy, how many labeled data and how many pre-determined pairwise labels are required for a good or perfect recovery of the label matrix $Z$.

Before stating the theoretical results, we first introduce some notations. Let $p_i$ denote the percentage of all examples in $\mathcal{D}_n$ that belongs to the $i$-th class. To facilitate our presentation and analysis, we also introduce a coherence measure $\mu_s$ of the orthogonal matrix $U_s = (\mathbf{u}_1, \cdots, \mathbf{u}_s) \in \mathbb{R}^{n \times s}$ as defined by

$$\mu_s = \frac{n}{s} \max_{1 \leq i \leq n} \sum_{j=1}^{s} U_{ij}^2 \tag{6}$$

The coherence measure has been exploited in many studies of matrix completion [29, 26], matrix reconstruction [23, 10]. It is notable that [4] defined a coherence measure of a complete orthogonal matrix $U = (\mathbf{u}_1, \cdots, \mathbf{u}_n) \in \mathbb{R}^{n \times n}$ by $\mu = \sqrt{n} \max_{1 \leq i \leq n, 1 \leq j \leq n} |U_{ij}|$. It is not difficult to see that $\mu_s \leq \mu^2 \leq n$. The coherence measure in (6) is also known as the largest statistical leverage score. Drineas et al. [8] proposed a fast approximation algorithm to compute the coherence of an arbitrary matrix. Intuitively, the coherence measures the degree to which the eigenvectors in $U_s$ or $U$ are correlated with the canonical bases. The purpose of introducing the coherence measure is to quantify how large the sampled labeled examples $m$ is in order to guarantee the sub-matrix $\widehat{U}_s \in \mathbb{R}^{m \times s}$ has full column rank. We defer the detailed statement to the supplementary material.

We begin with the recovery of the sub-matrix $\widehat{Z}$. The theorem below states if the the distribution of the data over the $r$ hidden classes is not skewed, then an $\Omega(r^2 m \log^2 m)$ number of pairwise labels between the labeled examples is enough for a perfect recovery of the sub-matrix $\widehat{Z}$.

**Theorem 1.** *Suppose the entries at $(i, j) \in \Sigma$ are sampled uniformly at random from $[m] \times [m]$, and the examples in $\widehat{\mathcal{D}}_m$ are sampled uniformly at random from $\mathcal{D}_n$. Then with a probability at least $1 - \sum_{i=1}^{r} \exp(-mp_i/8) - 2m^{-2}$, $\widehat{Z}$ is the unique solution to (2) if $|\Sigma| \geq \left\lceil \frac{512}{\min\limits_{1 \leq i \leq r} p_i^2} \right\rceil m \log^2(2m)$.*

Next, we present a theorem stating that if the column space of $Z$ is spanned by the orthogonal vectors $\mathbf{u}_1, \cdots, \mathbf{u}_s$ and $m \geq \Omega(\mu_s s \ln(m^2 s))$, the estimated matrix $Z'$ is equal to the underlying true matrix $Z$.

**Theorem 2.** *Suppose the entries at $(i, j) \in \Sigma$ are sampled uniformly at random from $[m] \times [m]$, and the objects in $\widehat{\mathcal{D}}_m$ are sampled uniformly at random from $\mathcal{D}_n$. If the column space of $Z$ is spanned by $\mathbf{u}_1, \cdots, \mathbf{u}_s$, $m \geq 8\mu_s s \log(m^2 s)$, and $|\Sigma| \geq \left\lceil \frac{512}{\min\limits_{1 \leq i \leq r} p_i^2} \right\rceil m \log^2(2m)$, then with a probability at least $1 - \sum_{i=1}^{r} \exp\left(-mp_i/8\right) - 3m^{-2}$, we have $Z' = Z$, where $Z'$ is computed by (5).*

Similar to other matrix reconstruction algorithms [4, 29, 26, 23, 10], the theorem above indicates that a low coherence measure $\mu_s$ plays a pivotal role in the success of the proposed algorithm. Actually, several previous works [23, 11] as well as our experiments have studied the coherence measure of real data sets and demonstrated that it is not rare to have an incoherent similarity matrix, i.e., with a small coherence measure. We now consider a more realistic scenario where some of the column vectors of $Z$ do not lie in the subspace spanned by the top $s$ eigen-vectors of the similarity matrix. To quantify the gap between the column space of $Z$ and the top eigen-space of the pairwise similarity matrix, we define the following quantity $\varepsilon = \sum_{k=1}^{r} \|\mathbf{g}_k - P_{U_S}\mathbf{g}_k\|_2^2$, where $P_{U_s} = U_s U_s^\top$ is the projection matrix that projects a vector to the space spanned by the columns of $U_s$. The following theorem shows that if $\varepsilon$ is small, so is the solution $Z'$ given in (5).

**Theorem 3.** *Suppose the entries at $(i,j) \in \Sigma$ are sampled uniformly at random from $[m] \times [m]$, and the objects in $\widehat{\mathcal{D}}_m$ are sampled uniformly at random from $\mathcal{D}_n$. If the conditions on $m$ and $|\Sigma|$ in Theorem 2 are satisfied. , then, with a probability at least $1 - \sum_{i=1}^{r} \exp(-mp_i) - 3m^{-2}$, we have*

$$\|Z' - Z\|_F \le \varepsilon \left(1 + \frac{2n}{m} + \frac{2\sqrt{2}n}{\sqrt{m\varepsilon}}\right) \le O\left(\frac{n\varepsilon}{m} + \frac{n\sqrt{\varepsilon}}{\sqrt{m}}\right)$$

**Sketch of Proofs** Before ending this section, we present a sketch of proofs. The details are deferred to the supplementary material. The proof of Theorem 1 relies on a matrix completion theory by Recht [20], which can guarantee the perfect recovery of the low rank matrix $\widehat{Z}$ provided the number of observed entries is sufficiently enough. The key to the proof is to show that the coherence measure of the sub-matrix $\widehat{Z}$ is bounded using the concentration inequality. To prove Theorem 2, we resort to convex optimization theory and Lemma 1 in [10], which shows that the sub-sampled matrix $\widehat{U}_s \in \mathbb{R}^{m \times s}$ has a full column rank if $m \ge \Omega(\mu_s s \log(s))$. Since $Z = U_s \left(\sum_{k=1}^{\top} \mathbf{a}_k \mathbf{a}_k^\top\right) U_s^\top$ and $Z' = U_s \left(\sum_{k=1}^{\top} \widehat{\mathbf{a}}_k \widehat{\mathbf{a}}_k^\top\right) U_s^\top$, therefore to prove $Z' = Z$ is equivalent to show $\widehat{\mathbf{a}}_k = \mathbf{a}_k, k \in [r]$, i.e., $\mathbf{a}_k, k \in [r]$ are the unique minimizers of problems in (4). It is sufficient to show the optimization problems in (4) are strictly convex, which follows immediately from that $\widehat{U}_s^\top \widehat{U}_s$ is a full rank PSD matrix with a high probability. The proof of Theorem 3 is more involved. The crux of the proof is to consider $\mathbf{g}_k = \mathbf{g}_k^\perp + \mathbf{g}_k^\parallel$, where $\mathbf{g}_k^\parallel = P_{U_s}\mathbf{g}_k$ is the orthogonal projection of $\mathbf{g}_k$ into the subspace spanned by $\mathbf{u}_1, \ldots, \mathbf{u}_s$ and $\mathbf{g}_k^\perp = \mathbf{g}_k - \mathbf{g}_k^\parallel$, and then bound $\|Z - Z'\|_F \le \|Z - Z_*\|_F + \|Z' - Z_*\|_F$, where $Z_* = \sum_k \mathbf{g}_k^{\parallel\top} \mathbf{g}_k^\parallel$.

## 5 Experimental Results

In this section, we present an empirical evaluation of our proposed simple algorithm for Transductive Pairwise Classification by Matrix Completion (TPCMC for short) on one synthetic data set and three real-world data sets.

### 5.1 Synthetic Data

We first generate a synthetic data set of 1000 examples evenly distributed over 4 classes, each of which contains 250 data points. Then we generate a pairwise similarity matrix $S$ by first constructing a pairwise label matrix $Z \in \{0,1\}^{1000 \times 1000}$, and then adding a noise term $\delta_{ij}$ to $Z_{ij}$ where $\delta_{ij} \in (0, 0.5)$ follows a uniform distribution. We use $S$ as the input pairwise similarity matrix of our proposed algorithm. The coherence measure of the top eigen-vectors of $S$ is a small value as shown in Figure 1. According to the random perturbation matrix theory [22], the top eigen-space of $S$ is close to the column space of the label matrix $Z$. We choose $s = 20$, which yields roughly $\mu_s = 2$. We randomly select $m = 4s\mu_s = 160$ data to form $\widehat{\mathcal{D}}_m$, out of with $|\Sigma| = 2mr^2 = 5120$ entries of the $160 \times 160$ sub-matrix are fed into the algorithm. In other words, roughly $0.5\%$ entries out of the whole pairwise label matrix $Z \in \{0,1\}^{1000 \times 1000}$ are observed. We show the ground-truth pairwise label matrix, the similarity matrix and the estimated label matrix in Figure 1, which clearly demonstrates that the recovered label matrix is more accurate than the perturbed similarities.

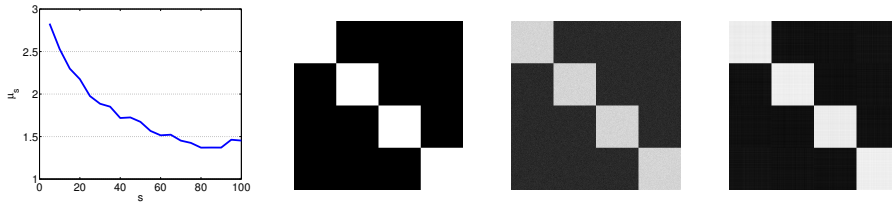

Figure 1: from left to right: $\mu_s$ vs $s$, the true pairwise label matrix, the perturbed similarity matrix, the recovered pairwise label matrix. The error of the estimated matrix is reduced by two times $\|Z - Z'\|_F / \|Z - S\|_F = 0.5$.

## 5.2 Real Data

We further evaluate the performance of our algorithm on three real-world data sets: splice [24] [3], gisette [12] [4] and citeseer [21] [5]. The splice is a DNA sequence data set for recognizing the splice junctions. The gisette is a perturbed image data for handwritten digit recognition, which is originally constructed for feature selection. The citeseer is a paper citation data, which has been used for link prediction. We emphasize that we do not intend these data sets to be comprehensive but instead to be illustrative case studies that are representative of a much wider range of applications. The statistics of the three data sets are summarized in Table 1. Given a data set of size $n$, we randomly choose $m = 20\%n, 30\%n, \ldots, 90\%n$ examples, where $10\%$ entries of the $m \times m$ label matrix are observed. We design the experiments in this way since according to Theorem 1, the number of observed entries $|\Sigma|$ increase as $m$ increases. For each given $m$, we repeat the experiments ten times with random selections and report the performance scores averaged over the ten trials. We construct a similarity matrix $S$ with each entry being equal to the cosine similarity of two examples based on their feature vectors. We set $s = 50$ in our algorithm and other algorithms as well. The corresponding coherence measures $\mu_s$ of the three data sets are shown in the last column of Table 1.

We compare with two state-of-the-art algorithms that utilize the pre-determined pairwise labels and the provided similarity matrix in different way (c.f. the discussion at the end of Section 3), i.e., Spectral Clustering (SC) [19] and Spectral Kernel Learning (SKL) [13] for the task of clustering. To attain a clustering from the proposed algorithm, we apply a similarity-based clustering algorithm to group the data into clusters based on the estimated label matrix. Here we use spectral clustering [19] for simplicity and fair comparison. For SC, to utilize the pre-determined pairwise labels we substitute the entries corresponding to the observed pairs by 1 if the two examples are known to be in the same class and 0 if the two examples are determined to belong to different classes. For SKL, we also apply the spectral clustering algorithm to cluster the data based on the learned kernel matrix. The comparison to SC and SKL can verify the effectiveness of the proposed algorithm for exploring the pre-determined labels and the provided similarities. After obtaining the clusters, we calculate three well-known metrics, namely normalized mutual information [9], pairwise F-measure [27] and accuracy [25] that measure the degree to which the obtained clusters match the groundtruth.

Figures 2~4 show the performance of different algorithms on the three data sets, respectively. First, the performance of all the three algorithms generally improves as the ratio of $m/n$ increases, which is consistent with our theoretical result in Theorem 3. Second, our proposed TPCMC performs the best on all the cases measured by all the three evaluation metrics, verifying its reliable performance. SKL generally performs better than SC, indicating that simply using the observed pairwise labels to directly modify the similarity matrix cannot fully utilize the label information. TPCMC is better than SKL meaning that the proposed algorithm is more effective in mining the knowledge from the pre-determined labels and the similarity matrix.

## 6 Conclusions

In this paper, we have presented a simple algorithm for transductive pairwise classification from pairwise similarities based on matrix completion and matrix products. The algorithm consists of two

Table 1: Statistics of the data sets

| name | # examples | # classes | coherence ($\mu_{50}$) |
|---|---|---|---|
| splice | 3175 | 2 | 1.97 |
| gisette | 7000 | 2 | 4.17 |
| citeseer | 3312 | 6 | 2.22 |

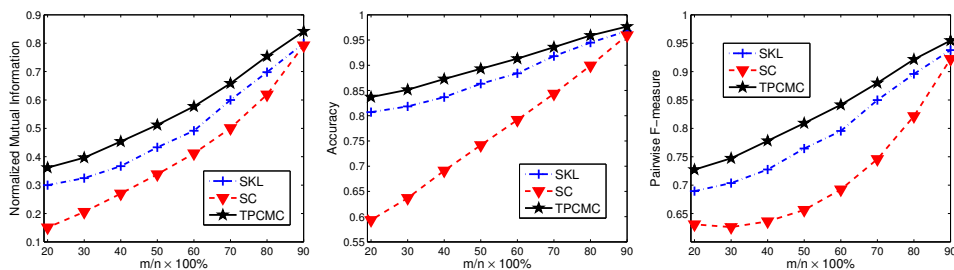

Figure 2: Performance on the splice data set.

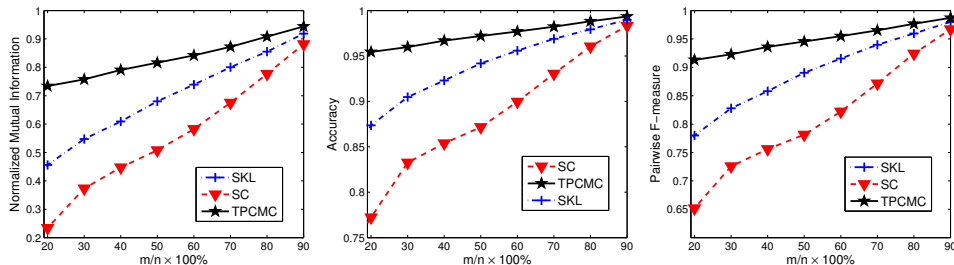

Figure 3: Performance on the gisette data set.

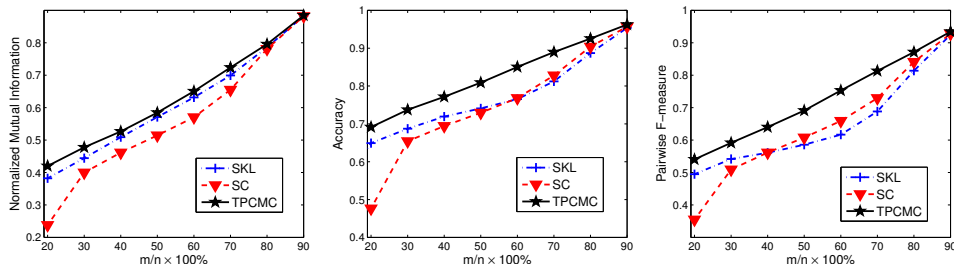

Figure 4: Performance on the citeseer data set.

simple steps: recovering the sub-matrix of pairwise labels given partially pre-determined pairwise labels and estimating the full label matrix from the recovered sub-matrix and the provided pairwise similarities. The theoretical analysis establishes the conditions on the similarity matrix, the number of labeled examples and the number of pre-determined pairwise labels under which the estimated pairwise label matrix by the proposed algorithm recovers the true one exactly or with a small error with an overwhelming probability. Preliminary empirical evaluations have verified the potential of the proposed algorithm.

## Ackowledgement

The work of Rong Jin was supported in part by National Science Foundation (IIS-1251031) and Office of Naval Research (N000141210431).

## Footnotes

[1]The pairwise similarities are usually derived from some side information instead of the underlying class labels.

[2]The labeled examples refer to examples in $\widehat{\mathcal{D}}_m$ that serve as the bed for the pre-determined pairwise labels.

[3] http://www.cs.toronto.edu/~delve/data/datasets.html

[4] http://www.nipsfsc.ecs.soton.ac.uk/datasets/

[5] http://www.cs.umd.edu/projects/linqs/projects/lbc/

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
