[Reviews · NeurIPS 2014]

Submitted by Assigned_Reviewer_11

The paper addresses the problem of pairwise classification. The authors propose a two-step algorithm that estimates if two data points belong to the same class. Additional to the method, they provide a theoretical analysis giving a bound for the minimum number of labels for the quality of the estimated labels.

The paper is well written. Related work seems adequate although I miss some references for pairwise classification. The algorithm seems partially novel yet simple. But the theoretical analysis lacks on a clear definition of the assumption, e.g., about the data distribution. Same information is also missing in the evaluation of the synthetic dataset. I guess the algorithm and the proof works only for a well-separated data distribution without overlapping cluster/classes?!

The results in Figure 2-4 are not really comparable in the direction of the x-axis as you increase with a larger m also the number of observed labels (10% of the mxm-matrix). It's not clear if the improvement comes from the increasing ratio m/n or only from the larger number of labelled data. Apart from that, there is no standard deviation. How large are differences between the runs?

Another question: I don't understand footnote 1. What is meant by side information?
Summary: This work addresses the interesting problem of pairwise labelling. Although the problem and the solution are derived naturally, there are essential informations missing to see the full benefit of the proposed method.

Submitted by Assigned_Reviewer_14

The authors study problem of transductive pairwise classification and propose a simple algorithm to address the task by using two step approach: 1) sub-matrix completion corresponding to the labeled examples 2) label matrix reconstruction from the provided similarity matrix and the completed sub-matrix. Furthermore, authors give theoretical result on reconstruction error and empirically evaluate performance of the proposed algorithm.

On one hand, it is certainly useful and interesting to consider transductive pairwise classification as a matrix completion problem. On the other hand, a natural setting for such task could be a semi-supervised or transductive preference learning. A large body of research exists on (ss) preference learning and ranking (e.g. work of Nir Ailon, Chu Wei, Johannes Fürnkranz, Eyke Hüllermeier, and others). These algorithms are frequently applied for link prediction, protein sequence ranking, etc, - basically all of the examples authors use to motivate their method. Why is this "setting" missing from the manuscript? Short description/reference to the relevant pairwise classification/preference learning algorithms as well as empirical comparison (e.g. to semi-supervised gaussian processes for preference learning) would further strengthen the paper.

The authors suggest notable differences of their approach (in particular transductive setting) compared to the algorithms presented in [25] and [23], however, a clear motivation for the proposed algorithm is lacking. While theoretical contribution of the paper is impressive, a related analysis has been described in [25] for semi-supervised setting and the strategy for perfect matrix recovery is quite similar to the one presented by the authors.

The experimental results section could be improved. For example, a comparison of several baseline methods on a synthetic data would be better than reporting the TPCMC result only. Also, it is not clear to me why "comparable" methods that can use side information (as in [25]) are not included in the evaluation. In fact, comparison is done only with two clustering techniques. Finally, performance of the methods such as SC or SKL greatly depends on a number of parameters, choice of kernel, they way Laplacian matrix is construed, etc. How were these parameters estimated? Were the experiments performed multiple times and the average results reported as in [25]? Unless some of these questions are addressed it is difficult to agree with the last statement in the manuscript, that is "empirical evaluations have verified the effectiveness of the proposed algorithm".
Summary: This is a nice contribution that not only describes a novel transductive pairwise classification algorithm but also provides theoretical insights on reconstruction error of the proposed method.

Submitted by Assigned_Reviewer_45

The paper studies transductive classification from pairwise similarities. The paper presents a two-step approach. The first one is to complete submatrix of pairwise similarity corresponding to the labeled examples. The second one is to reconstruct the label matrix from the completed sub-matrix and the provided similarity matrix. The paper shows theoretical guarantee for the proposed approach. That is, when top eigen-space of the similarity matrix covers well the column space of label matrix, and the number of observed pairwise labels is sufficiently enough and some mild preconditions, the proposed approach recovers the label matrix with a small error.

The paper is generally well written. The proposed approach is technically sound.

Followings are two comments.

1. According to theorem 1, the perfect recovery requires sufficient large number of pairwise labels. This might be difficult in practice. If the number of pairwise labels is not sufficiently enough, how will it affect the recovery of label matrix?
2. The proposed approach contains two steps. The error of the second step will be affected by the error of the first step. I am wondering how does the error of first step affect the error of second step?

>> comments after reading authors' response

Thanks for the response. It addresses most of my concerns. I would like to stick to my rating, i.e., a good paper.
Summary: The proposed approach is technically sound with informative theoretical analysis. Two minor comments about the proposed approach: a) How will the number of pairwise labels affect the recovery of label matrix? b) How does the error of the first step affect the error of the second one?
Author Feedback
Author rebuttal: We are grateful to all reviewers for their useful comments and suggestions.

Assigned_Reviewer_11:
Q: the theoretical analysis lacks on a clear definition of the assumption, e.g., about the data distribution.

A: As stated in the second paragraph of Section 4, “Let p_i denote the percentage
of all examples in D_n that belongs to the i-th class”. The theoretical analysis is built on the empirical distribution of $n$ data points over $r$ hidden classes.

Q: I guess the algorithm and the proof works only for a well-separated data distribution without overlapping cluster/classes?

A: The proof works for any data distribution. However, in skewed distributed cases, our theoretical results in Theorem 1 and Theorem 2 exhibit that it is more difficult to recover the full pair-wise label matrix than in well-separated cases, which is also consistent with previous studies that observed poor performance of spectral clustering and semi-supervised learning for unbalanced data.

Q: The results in Figure 2-4 are not really comparable in the direction of the x-axis as you increase with a larger m also the number of observed labels. Apart from that, there is no standard deviation. How large are differences between the runs?

A: The results in Figure 2-4 demonstrate that under the same number of observed entries and the same number of labeled data, the proposed algorithm does better in utilizing the provided information for estimating the pairwise labels. We increase the number of observed entries with $m$ due to Theorem 1. We would provide plots with standard deviation in the final version.

Q: What is meant by side information?

A: The side information is a vague term, used to refer any information rather than the class label information. For example, the pairwise similarity between any two images can be computed from visual features.

Assigned_Reviewer_14:
Q: Why is this "setting" (semi-supervised or transductive preference learning) missing from the manuscript?

A: We did not consider the setting of preference learning because that the ground-truth that we analyzed about a pair of data denoted by $h(u,v)$ is a symmetric function, $h(u,v)=h(v,u)$. In preference learning, the function $h(u,v)$ is an asymmetric function. We would add some discussions into the final version.

Q: While theoretical contribution of the paper is impressive, a related analysis has been described in [25] for semi-supervised setting and the strategy for perfect matrix recovery is quite similar to the one presented by the authors.

A: Although both works consider the semi-supervised setting, however, we consider a much more difficult setting, i.e., the transductive setting, and the underlying logic is different. The algorithm in [25] is a side information aided matrix completion using the provided similarities to regularize the matrix to be completed. In contrast, our algorithm is a two-step procedure. The first step is a standard matrix completion and the second step is a least square estimation using the provided similarities. Therefore our analysis is completely different from the analysis in [25]. Our analysis is based on the standard matrix completion theory and the convex optimization.

Q: It is not clear to me why "comparable" methods that can use side information (as in [25]) are not included in the evaluation.

A: We did not compare with [25] because its success relies a critical assumption that the observed entries are randomly distributed over [n]*[n] (all pairs of data), where n is the total number of data points. However, in our setting, the observed entries only randomly span over [m]*[m] (a small pairs of data).

Q: Finally, performance of the methods such as SC or SKL greatly depends on a number of parameters, choice of kernel, they way Laplacian matrix is construed, etc. How were these parameters estimated?

A: In the experiments on real data sets, we use the cosine similarity (no parameter involved). The Laplacian is constructed as given in the paper [16] (i.e., L = D^{-1/2}SD^{-1/2}. For SKL algorithm [12], the only parameter is $C$ the ratio between consecutive eigen-values. We set $C=2$ as observed to yield stable and good performance in [12].

Q: Were the experiments performed multiple times and the average results reported as in [25]?

A: As stated in the paper “For each given m, we repeat the experiments ten times with random selections and report the performance scores averaged over the ten trials.”

Assigned_Reviewer_45:
Q: According to theorem 1, the perfect recovery requires sufficient large number of pairwise labels. This might be difficult in practice. If the number of pairwise labels is not sufficiently enough, how will it affect the recovery of label matrix?

A: The Theorem 1 indicates that if the distribution of the data over the r hidden classes is not skewed, then only $O(r^2m\log^2m)$ (a linear order of $m$) pairwise labels are sufficient to recover the label matrix. On the other hand, the theorem implies that if the number of pairwise labels is not sufficient, it is difficult or even impossible to recover the full label matrix. The error propagation is discussed below.

Q: The proposed approach contains two steps. The error of the second step will be affected by the error of the first step. I am wondering how does the error of first step affect the error of second step?

A: Since the estimation of the label matrix is based on the estimation of \sum_k a_ka_k^T, where $a_k$ is the coefficient in $g_k = Us*a_k$. The first step can be considered to providing an estimation of $g_k$. Therefore the error in the first step will affect the accuracy of $g_k$ and consequently the accuracy of $a_k$--the solution to the linear system \min_{a_k} $|g_k – Us*a_k|^2$. Therefore previous error analysis for linear system can help us analyze the error propagation. We will add some discussions to our supplementary materials.